# BCI Applications to Creativity: Review and Future Directions, from little-c to C^2^

**DOI:** 10.3390/brainsci13040665

**Published:** 2023-04-15

**Authors:** Maria Elide Vanutelli, Marco Salvadore, Claudio Lucchiari

**Affiliations:** Department of Philosophy “Piero Martinetti”, Università degli Studi di Milano, 20122 Milan, Italy; marco.salvadore@studenti.unimi.it (M.S.);

**Keywords:** BCI, EEG, creativity, little-c, C^2^, hyperscanning, hyperfeedback, multi-brain, creative enactment, art

## Abstract

BCI devices are increasingly being used to create interactive interfaces between users and their own psychophysiological signals. Over the years, these systems have seen strong development as they can enable people with limited mobility to make certain decisions to alter their environment. Additionally, their portability and ease of use have allowed a field of research to flourish for the study of cognitive and emotional processes in natural settings. The study of creativity, especially little creativity (little-c), is one example, although the results of this cutting-edge research are often poorly systematized. The purpose of the present paper, therefore, was to conduct a scoping review to describe and systematize the various studies that have been conducted on the application potential of BCI to the field of creativity. Twenty-two papers were selected that collect information on different aspects of creativity, including clinical applications; art experience in settings with high ecological validity; BCI for creative content creation, and participants’ engagement. Critical issues and potentialities of this promising area of study are also presented. Implications for future developments towards multi-brain creativity settings and C^2^ are discussed.

## 1. Introduction

### 1.1. Brain–Computer Interface

An electroencephalogram-based (EEG) brain–computer interface (BCI) is a headset that allows one to detect, record, and analyze cerebral activity in real time and use it to interact with a computer for some purposes. Most modern BCIs are low-cost, highly portable, and non-invasive. As they are wireless, they allow people to feel free from constraints and to move freely. This makes it easier to run complex tasks, such as those required in creative settings. In addition to frequency, a BCI can analyze temporal features (e.g., event-related potentials (ERPs)). Most BCI systems come with software that processes the data stream and provides values related to the mental state of the user. Relaxation and arousal indices are generally computed online, and they can be used directly to perform a task, control a digital interface, or set an analog device. However, raw data are available for some devices so that ad hoc EEG computations can be run as well. In this way, a BCI is a very flexible tool that opens up an indefinite number of applications, experiments, and practical uses. Furthermore, it allows researchers to run natural or real-world studies. Another possible use of a BCI is in neurofeedback, which is the modulation of cerebral activity during a task based on certain feedback about the ongoing EEG. It can be used, for example, to improve focal attention [1], working memory [2], emotional management [3], learning [4], and creative thinking [5] in both experimental and real-life contexts. In addition, multiple settings can be implemented. In this application, two or more people wearing a BCI cooperated to perform a certain task. Cooperative BCI paradigms have been specifically designed to improve the joint performance of users [6]. They are made up of three main parts: a data recording module, a signal elaboration module, and a command translation module. Consequently, there are three important procedures. First, the different members’ brain signals are acquired by multiple BCI recording devices and are then synchronized through the use of triggers. Second, phase-locked data are elaborated and integrated by extracting indices to decode the users’ mental states. Third, after the indices are extracted, they are converted directly into commands, which can be used for various purposes, for example, to regulate emotions, to paint on a screen, or to provide feedback. For the latter case, we previously defined the concept of hyperfeedback [7] as a new paradigm to implement multi-brain neurofeedback settings. Compared with a single BCI setup, the complexity of the multi-user input system brings about technical challenges for both the recording procedures and the signal processing. However, its potential for use in ecological and interactive contexts could provide interesting insights for various research protocols, especially in studies of creativity.

### 1.2. Creativity: Some Definitions

When facing the creativity problem, we need to consider it as a metaphoric concept [8]. This means that “creativity” is not only a common term used to indicate something novel or innovative. Creativity is a mental crossroad at which several concepts, intentions, and emotions meet. Extending the metaphor, we could say that creativity is also a bridge between scientific disciplines, research methods, and technology. It might also be considered a “mental state” in which many pathways converge and from which original ideas may arise. If so, it is clear that it is impossible to find just one definition. However, we can mention at least some core components of creativity that most research considers fundamental to the study of the so-called little-c, that is, everyday life creativity [9]. The basic components are divergent thinking, convergent thinking, flexible cognition, and enactment.

Divergent thinking is often referred to as ideation: the ability to create new ideas, starting from an input. Children seem to develop this ability around 2 years of age [10]. Divergent thinking is the opposite of functional fixity, which occurs when a routinized way of thinking inhibits new ideas, leaving the thinker with just a few ideas linked to previously tested solutions.

Divergent thinking is a good predictor of creativity; it is generally better than intelligence, as measured by IQ tests [11], but it is not the full creativity. It is an important aspect, mainly based on fluency, that is the ability to generate new solutions to new or ambiguous problems in a short time. However, possessing fast and fluent thinking is not a guarantee of creativity. Indeed, creative people must find an efficient way to integrate information to give rise to a useful outcome, which is achieved thanks to convergent thinking. From a neurological perspective, this mechanism relies on two main processes: the first is the spread of neural activation, and the second is the new connection of activated areas that were previously unconnected [12]. Divergent and convergent thinking are often considered two different mindsets, metaphorically described as the artist and the guard. Although flexible cognition, which is the capacity to see things differently, is usually referred to as divergence, it can be associated with both mindsets. In truth, the capacity to switch between the two and choose the best one based on the context is probably the best way to picture flexible cognition [12].

Finally, a fundamental aspect of creativity we want to introduce is enactment. When someone gives rise to a mental product, they must transfer it to the external world through their body, actions, language, and so on. In this phase, creativity must be enacted. Imagine a songwriter who has thought of a sound or text. In their mind, the final outcome is probably the result of a number of steps, beginning from inspiration and divergent thinking and finishing with editing (convergent thinking), finalized by executive functions. However, the real outcome of writing is not just a piece of paper made by rows and signs (words or notes) but a global result of a complex process. To convey the intended message, the author combined those signs into a specific shape, which allowed for their “revitalization”. This process is what we call enactment. For instance, a singer must find a way to get into the song and go through the creative process that produced it. This is another act of creativity, but the starting point is exactly the endpoint of the previous act of creativity. This aspect was scientifically investigated by Charles Limb [13], who studied the dynamics of the brain during improvisation by jazz musicians. In this case, people are not simply giving rise to new ideas: they are generating thoughts, movements, and sounds to enact a performance, which may be artistic, ecstatic, or even intellectual. This is due to a particular neuro-cognitive pattern that weakens the performer’s self-control, probably because of hypofrontality, the partial shutdown of the pre-frontal cortices [14,15], which allows for enaction without the supervision of the prefrontal cognitive control network. We argue that a BCI-based device is particularly useful for studying enactment, especially in natural settings.

### 1.3. The Role of BCI in the Study of Creativity

The study of creativity using BCI systems can adopt classical paradigms used in EEG studies. However, the specific characteristics of a BCI have generally led researchers to take advantage of the possibility of running ecological studies and/or to combine psychological experiments, technology, and arts to collect data that are not easily accessible using other methods and techniques. Studies on divergent or convergent thinking are not frequent since they are generally conducted using a traditional EEG or other neuroscientific methods. Instead, many studies focusing on enactment have been conducted employing BCI. Furthermore, BCI is considered a frontier in the development of new art forms and the implementation of creative and dynamic human–machine interfaces and in giving rise to multi-brain settings designed to enact or improve collective creativity. In this framework, clinical applications to assist patients’ creative expression are also important. We believe that research on creativity must also be developed in real-world settings and that in a near future, more BCI studies will target divergent and convergent thinking to fill the present gap between laboratories and natural studies.

In the following sections, we describe the methodology we used to select and review studies to investigate how BCIs are used in research on creativity. Finally, we propose future scenarios in the development of BCI applications to further investigate creativity and the implementation of tools that are able to extend its borders in scientific and applied contexts.

## 2. Materials and Methods

The present work was conducted based on the Preferred Reporting Items for Systematic Reviews and Meta-Analysis Extension for Scoping Reviews (PRISMA-ScR) [16]. In addition, to better map and describe the review process, we followed the-five stage framework by Arksey and O’Malley for a scoping review [17]. The five stages were pursued following an iterative process consisting of (1) identifying the research question and (2) relevant studies; (3) study selection; (4) charting the data; (5) collating, summarizing, and reporting the results.

### 2.1. Stage 1: Identifying the Research Question

The question that gave rise to the present research was if and how BCI can provide insight into the creative process.

### 2.2. Stage 2: Identifying Relevant Studies

To answer this question, we adopted a wide definition of creativity, including both divergent and convergent thinking applied to a variety of different tasks and activities, such as dance, music production, and the arts in general, i.e., different forms of enactment.

To identify the pertinent papers for screening, we used the following keywords: ((“BCI” OR “brain–computer interface”) AND (“creativity” OR “divergent thinking” OR “convergent thinking” OR “art” OR “dance” OR “cinema” OR “theatre” OR “theater” OR “music”)).

The search was conducted within the PubMed and Scopus databases. The results were updated until 13 January 2023. As the BCI is quite a recent methodology, we did not place any time limit on the search.

We then applied the following criteria to the results to select the pertinent papers for screening. The inclusion criteria were: (1) papers written in English; (2) studies describing innovative methods to facilitate creative expression; (3) studies using a wearable EEG-BCI; The exclusion criteria were: (1) reviews, metanalyses, and book chapters; (2) studies aimed at validating protocols or interfaces without addressing the creative process; (3) studies using BCI systems that are not wearable. Considering that this is an innovative and cutting-edge topic, we decided not to discard research protocols or conference papers that focused on creative processes.

### 2.3. Stage 3: Study Selection

M.E.V. and C.L. each screened the relevant articles by title, keywords, and language and then by reading the abstracts and full texts. The selection of studies followed the Preferred Reporting of Items for Systematic Reviews and Meta-Analyses (PRISMA) Statement [18] (see Figure 1). In cases of uncertainty about including or discarding papers, M.E.V. and C.L. held a discussion to reach a consensus. After the selection of the relevant studies, the references were screened to include other possible pertinent studies.

### 2.4. Stage 4: Charting the Data

A table was produced to compare the selected papers. Where possible, the extracted data followed the recommendations of Arksey and O’Malley [17]. Each row reports the authors and year of publication, the country where the study was conducted, the size of the sample, the experimental framework for creativity, the assessment tools, BCI setup, and the main results (see Table 1).

### 2.5. Stage 5: Collating, Summarizing, and Reporting the Results

An analytic framework was considered to present a narrative report of the existing literature, following the PRISMA guidelines extension for scoping review [16].

## 3. Results

### 3.1. Characteristics of the Studies

A total of 22 studies were selected. We subdivided them into four categories based on the type of creativity they considered: clinical applications, art experience in naturalistic settings, content creation, and participants’ engagement (see Table 1). The clinical applications section describes studies specifically designed to allow users with motor impairment to express their creativity by implementing interactive digital interfaces to be controlled by different electrophysiological markers. The section on art experience in ecological settings proposes innovative systems that can be used in real-life creative contexts, such as museums. The content creation section describes the selected studies that explored proto-creative compositions through interfaces that the users could interact with to select elements to be composed together. The articles focused on both musical and figurative expression, as well as the production of hybrid creative artifacts which are generated by the interaction between the user’s creativity and the interfaces. The last section contains all contributions that addressed the idea of using participants’ engagement to create and modify interactive artifacts.

### 3.2. Clinical Applications

Levican and colleagues [19] adopted and tested a very interesting approach that combined a BCI device and a creative task in a clinical setting. In particular, the authors recorded EEG data of a patient with locked-in syndrome and used online streaming to allow the person to produce music or at least modulate it. The researchers developed their own software to process raw data and used the synchronization of an adapted alpha rhythm (3–7 Hz) to control volume, while data from the head-in accelerometer were used to modulate tempo and harmony. Furthermore, the application of an event-related approach was also integrated to improve their system. In particular, to distinguish the baseline values from task-related values, they asked the person to perform mental arithmetic. Thus, when the user began counting, the system recorded an increase in the ERP values and could use this datum as a marker to change some parameters of the sound or as an on/off switch. Tested with Alberto Vega, an actor with locked-in syndrome, the system showed promising results.

An older study by Münßinger and colleagues [20] tested the feasibility and reliability of a BCI system designed to allow users to use a painting application. The authors recruited ten healthy participants and three people with amyotrophic lateral sclerosis (ALS) and asked them to draw on the screen using their EEG signals (brain painting) by using an ERP paradigm based on the P300 component. The study also included psychological scales to measure mood, feasibility, and emotional reactions. Since the sample of patients with ALS was small, the study reported statistical data on only the healthy participants, who demonstrated an ability to learn how to use the device in a fairly reliable fashion and with an acceptable effort. The patients also showed that the system could be considered an interesting tool for allowing people with severe neurological conditions to express their creativity. It could be also used to involve patients in engaging tasks that could relieve stress and negative psychological emotions. However, the task was much more difficult for the patients than the healthy (and younger) participants, and a special adjustment of the tool should be adopted to increase the reliability of the system and its effectiveness and to reduce time and cognitive resources.

A third contribution is very early, yet interesting. In his paper, Miranda [21] proposed an innovative BCI system that not only reads the EEG of the user but also activates generative rules based on the frequency bands. Tempo and loudness modulation can also be performed based on signal complexity. The users can then also learn to select among different generative rules to freely express their creativity.

### 3.3. Art Experience in Real-World Settings

A first contribution by a Canadian research group [22] described a prototype called iMind that allows the user to select pieces of art by Paul Klee (digitalized versions) that most mirror their current emotional state. In this way, the authors created a bridge between the artworks and the observers. The basic idea is that the collection of EEG data from a person observing a digital artifact may be used to generate a dialogue between the artistic offline coding and the ongoing neural coding of the observers. The authors also suggested a possible dual-user application (emotional dialogue) and proposed this project as means of bringing the public closer to the authors’ work.

Herrera-Arcos and colleagues [23] conducted one of the few controlled experimental studies in this area during a museum tour. While wearing a BCI, the participants visited a museum and then selected their most appreciated piece of art. After collecting the participants’ online data, the researchers found that the suppression of beta-band frequencies over the frontal electrodes may be considered a marker of the experience of viewing one’s preferred painting. With this study, the authors underlined the importance of considering the subjective experience during art consumption, which is untied from the artist’s style and intentions.

These studies, though not directly linked to creative processes, are important for analyzing the neuro-cognitive effects creative products have on people. In this way, thanks to the use of methods with high ecological validity, it is possible to understand how creativity may specifically affect observers with different characteristics and to potentially use this data to support creative processes in school or the workplace.

### 3.4. Content Creation

#### 3.4.1. Proto-Creativity

This section includes five papers describing the use of a BCI for very simple creative compositions in which the user can interact with an interface to select elements to be composed together. We considered these studies investigations of proto-creativity since they analyzed the feasibility of their devices, which created a basic system to allow a person to express their creativity. However, one study did not report data about the use of the developed systems to give rise to full creative products. Two papers describe painting applications, while the other three are about music composition. The two papers describing painting applications were published 10 years apart. The first paper [24] explored the possible creative applications of a BCI by using steady-state visual-evoked potentials (SSVEPs) in combination with different interface matrices. The second paper [25] used an SSVEP in combination with P300. Both studies applied the interface to help the participants create different drawings, either by copying or through free expression.

Moving forward, a similar principle was applied to music composition. In detail, Hamadicharef and colleagues [26] and Pinegger and colleagues [27] presented a P300-based BCI to select individual musical notes for composing melodies by choosing from a matrix. This application could be further developed to be used for content creation (see the next paragraph) if accompanied by the users’ intention to actively modulate their brain activation to manipulate music composition. Finally, Vamvakousis and Ramirez [28] used a P300 to change the harmony of an arpeggio. In this last case, the application was even more interactive in that the user could voluntarily influence music while listening to it.

#### 3.4.2. Creative Artifacts Production

In this section, six papers are presented that describe an interactive creative expression by users through a BCI. The most recent paper was authored by Riccio and colleagues [29] and addressed the creation of paintings. In this case, the authors implemented an interface based on a generative adversary network (GAN) that generated a painting based on the imputed EEG signal. Of course, this procedure was made possible after a preliminary learning phase in which different EEG sections (the first database) were associated with different emotions while viewing a second dataset of paintings. The emotional labels associated with the elements of the two databases functioned as a feed for the GAN to learn how to associate these patterns. The participants were then asked to rate the new, original paintings based on the emotional categories.

A Korean study [30] explored the possibility of creating self-portraits based on the information conveyed by the BCI system. In detail, the color, thickness, and sharpness of the lines changed based on the cortical activity of the participants. For example, an anxious user would have a red-nuanced portrait with very thick lines. The authors proposed this method as a way to remind people to look after their mental well-being.

Folgieri and Zichella [31] used a combination of EEG data, visual stimulation, and motor gestures to compose music. They also transformed the EEG spectra of each participant into a melody by converting online the prevalence of one band over the other, thus obtaining personalized music.

In a later work, Folgieri and colleagues [32] presented an interface to be used in both the controlled conditions of a lab, to explore the cognitive and emotional correlates of creativity, and in live, hybrid artistic performance. Their system, called DRACLE, collected live EEG data from participants and returned the data through a Blender graphic involving both visual (a spiral of spheres) and auditory (volume adjustments) feedback. The data suggested that people implicitly learn how to adjust their EEG to obtain a satisfying outcome and reported positive emotional experiences (see Figure 2).

Additionally, Cádiz and de la Cuadra [33] presented KARA, their project to be applied during real-life music performances. The innovative aspect of KARA is that while the users play the cello (KARA 1) or the flute (KARA 2), they wear a BCI that translates their brain activity into a visual pattern that accompanies the performance. In this way, participants give rise to a complex artistic product made by conscious and unconscious processes, potentially with powerful effects on observers.

Finally, the enactive mandala is an example of a digital art tool developed by Lyons and Tokunaga [34]. Participants interact with a digital interface that displays a cloud of particles that can be transformed into a coronal figure when the EEG data collected mark an increase in relaxation. In this way, participants attempt to change the order and the shape of the particles displayed by changing their mental state. The tool aims to study how a person can translate feelings and moods into visual graphics and sounds, also in the context of artistic performance, and to analyze how the enactive mandala may support significant human–computer interactions for future applications (see Figure 3).

### 3.5. Participants’ Engagement

One of the relevant means of using a BCI in a creative context is to measure individual EEG indices related to attention and engagement. One of the main characteristics of a creative context is inspiration. To be inspiring, a lesson, a discussion, or a performance must be able to engage participants. We found four articles describing studies in which a BCI was used for this aim. In particular, the work by Yan and colleagues [35] used a passive BCI approach to measure participants’ engagement. The authors also developed an active interface that was able to react to a change in audience engagement. During two studies, they collected data in an individual setting. The participants were immersed in two different performances, which were shown on a semi-circular screen. Using the ratio between beta and alpha and theta as an engagement index [41], individual thresholds were established to evaluate when a person was at a certain engagement level, and specific cues were provided by a digital interface to increase engagement when needed. Subjective data were also collected through Likert scales. The data showed that providing cues aimed at increasing engagement in low-level periods increased both objective and subjective measures. In particular, the cued conditions increased the participants’ ability to remember facts after the performance, and the participants also reported being more involved in the narration when the level of engagement was kept high. This study suggests that a BCI device coupled with a dynamic interface can be used to monitor and manage participants’ engagement in passive creative contexts, i.e., observing artistic performances. However, the results suggest that a similar system might be useful in active creative contexts as well, for example, in groups involved in problem-solving and in learning contexts.

The approach used by Ramchurn and colleagues [36] is particularly interesting. They implemented a BCI-based system to allow a person to edit a movie instead of controlling the narration. In this way, they attempted to empower the observers by providing them with the possibility of unconsciously editing the movie, for instance, cutting a scene when the participant’s level of attention/engagement dropped. They also designed a practice-led experience in a naturalistic ecological setting in which one controller used the BCI-based system and five true observers watched an ad hoc controlled movie. They collected data both from questionnaires and interviews, and their results suggested that the BCI-led cut system was appreciated and favored the audience’s engagement and ability to remember the story as well as single scenes. The same system has been used to adapt the soundtracks of particular scenes to the level of engagement of participants. In this way, it was possible to provide musical stimuli that were coherent with the narrative and specifically designed to maintain the observer’s engagement with the story [37].

Marchesi et al. [38] realized an interactive cinema experience based on a narrative that can be modulated by the EEG participants, who are measured by a BCI device. Using a Blender script, the authors proposed a short movie made with an introduction and two possible ends. The EEG activity of the participants was used to opt for the best end for each participant.

Later, a study from Marchesi [39] explored the possibility of interacting with a movie by marking the sequences according to the mood of the user, as assessed by the BCI. In this way, when the story is played, the users’ reactions can affect the development of the narration. The study did not present data; instead, it described possible future applications in interactive cinema.

Finally, the work of Zioga and colleagues [40] applied BCI devices to three live performances in a multi-brain setting. In each condition, one performer (an actress) and two people from the audience wore the BCI. Interestingly, the brain activity recorded was not only used to detect the level of engagement but was also part of the creative process itself. Initially, the brain activity of the actress was used to change the colored filter that was applied to the video stream. In the second phase, the brain activity of the audience was used with the same aim. Finally, the brain activities of the performer and the audience were merged to obtain an interactive and dynamic live performance and to produce an immersive dramaturgical experience. Objective and subjective data showed that the use of the BCI allowed one to obtain a deep level of engagement. People were generally able to recognize when the brain activity, and therefore their experience, was having an impact on the scene. Interestingly, the EEG gamma band (25–40 Hz) increased significantly when the performance was modulated by the interaction of the actress and the audience’s performance, likely indicating the effect of full engagement in the ongoing dramaturgical narration. Finally, a subsequent memory assessment showed that the best-remembered parts of the performance were the ones marked as more engaging by the EEG data.

## 4. Discussion, Conclusions, and Future Directions

### 4.1. Highlights from the Scoping Review

Creativity can be studied through a multitude of research methods and data collection techniques. However, classic studies succeed in capturing a kind of creativity that is static, fragmented, and reduced to a few of its components. The dynamic and associative nature of creativity is mostly lost in favor of rigorous yet cold methodologies and settings [42].

A BCI makes it possible to design and implement studies with a much higher ecological validity than traditional studies. Creativity can thus be studied not only as a cognitive or neuro-physiological process but also as a dynamic, socio-cognitive process that is one of its primary components. Indeed, it is possible to place participants in experimental situations in which they are free to fully express their creativity. In addition, a BCI makes it possible to implement peculiar forms of creativity. First, it enables people who cannot express their creativity due to neurological problems affecting speech, gesture, and mobility by taking advantage of the possibilities provided by computers and artificial intelligence. Through training, which is generally not too demanding, people can learn to interact directly with a computer to express their graphic and musical creativity [20]. However, the development of increasingly sophisticated artificial intelligence engines and the ability to turn text into images and vice versa [43] provides multiple advantages/opens various doors. First, it opens the door for linguistic and sophisticated types of creativity that call upon the construction of stories and multimedia scenarios. Therefore, BCIs and artificial intelligence represent a fundamental frontier to all those who can no longer express creativity in traditional ways due to physical or mental reasons. Since the relationship between creativity and psychological well-being is well-known [44], many applications may soon be developed to improve the quality of life of various categories of patients. In addition, such applications may also be used by healthy people to experiment with new forms of creative modalities to increase their creative potential and enactment or to develop specific cognitive skills.

A second area that we believe is crucially important in the study of BCI-mediated creativity is hybrid creativity. Several research studies have already shown how BCIs and the development of dedicated software can help people express forms of creativity that cannot refer exclusively to either a human brain or a computer [44,45,46]. It is the interaction between the two data processors (the brain and the computer), and thus between natural and artificial intelligence, that brings creative products to life. It is a circular process in which the computer interprets data from an individual’s brain and transforms it into something creative, such as an iridescent graphic form. At the same time, the users observe the computer-produced change in the visualization and try to modify it to their liking, thus prompting the computer to modify the graphics in a circular and potentially infinite loop. Since it is impossible to tell where the generation of creativity lies, this process cannot be referred to as a single-actor process. In addition to allowing for an intense and original creative experience, the system also allows for the collection of data on implicit learning and the interplay between conscious and unconscious processes. Finally, it could be used as a tool to increase attentional focus [1], to support learning processes [4], and as a creative warm-up [5,7].

A third area of innovation concerns the implementation of multi-brain interactive studies and devices. Neuroscience is already moving in a multi-brain direction by expanding the classic individual paradigm to include the study of multiple people simultaneously engaged in a certain task, and in particular, to investigate how those brains—directly and indirectly—modulate each other. The human brain rarely works in isolation and is always included in a physical environment and a social environment. Indeed, research strongly suggests that reasoning, deciding, solving a problem, or even simply listening to music or watching a movie alone are not the same as engaging in these activities with other people [45,47]. The brain works differently, and neuroscience that only knows how to study brains in individual settings would be a limited science in terms of its ability to produce knowledge [48]. The hyperscanning paradigm thus originated as a study of collaborative or competitive contexts, but it can be extended to a more general understanding of the social dimension of cognition. In this sense, BCIs represent a privileged tool, as they allow data to be collected from two or more people in naturalistic ecological contexts. Recently, we proposed hyperfeedback [7], which has all the advantages of neurofeedback yet allows for implicit socio-cognitive learning to be generated by exploiting implicit and automatic collaboration and synchronization between people involved in the same group setting. In this sense, a BCI can be a particularly flexible tool, as hyperfeedback could be carried out under conditions of extreme freedom flexibility. Moreover, it allows for the easy implementation of flexible creative settings and avoids the so-called creativity killers [48] that often are present in standardized and rigid experimental settings.

Some studies have already investigated the possibility of using multiple BCI devices simultaneously within a theatrical performance to connect spectators with performers [35,40]. This process results in the modulation of the creative performance by processing implicit signals from multiple people whose brains somehow connect truly, rather than virtually, to the immersive creative experience. The creation of a shared creative space fosters engagement and even the recollection of narrative events. Thus, it demonstrates the potential of such methodologies to not only enhance the creative experience but also to facilitate the implicit learning processes that are so vital yet difficult to address. Moreover, the group setting constitutes the classic learning setting in schools. Therefore, collecting data about the connection between different learners’ brains and using this data to provide feedback and modulate the class contents can be an effective and inexpensive way to enhance the learning process of the whole class group in both creative and noncreative settings.

### 4.2. Future Directions and Perspectives

Nevertheless, we think there are other and much broader horizons that remain to be explored in the domains of creativity and multi-brain research. Indeed, creative processes are usually studied in individual settings since creativity is considered a personal route. However, several authors have proposed concepts of non-individual creativity, such as group creativity or class creativity [49]. These are forms of creativity that cannot be referred to a single individual and therefore must be measured in terms of the change in a group’s performance. For example, when studying class creativity, the target is not the single individual but the class as a whole. All data collected are thus to be referred exclusively to the group. Similarly, class creative training is designed and implemented to foster exchange, interaction, the clash of positions, and the emergence of discussed and shared innovations. Metaphorically speaking, we could say that the goal of implementing collective creativity allows one individual in the group to conclude another’s story in a way that satisfies both in terms of the final product and of the (implicit) collaboration. In practice, a spontaneous process is established that leads each participant to close out the other’s thought in such a natural way that it is perceived not as an external addition but as an internal and self-related process, as if it were one’s own thought.

All these processes can be contextualized within the framework of the extended mind, which encourages considering the environment (in this case, a social one) at the same level as the cognitive and internal processes [50]. Achieving this result is very challenging and is generally implemented in restricted contexts. From our perspective, the use of BCI can accelerate this process, particularly in dyadic contexts.

Indeed, the use of a large group does generate greater richness on the level of a variety of ideas, visions, and perspectives and thus in terms of divergent thinking. However, it may also generate confusion on both cognitive and psycho-social levels. Such complexity makes the process long and cumbersome, and the results achieved are often transient. In addition, the group is often a hindrance to the level of convergent thinking, editing processes, and systematization of ideas. A dual setting, on the other hand, makes it possible to take advantage of the cognitive extension provided by the comparison of different perspectives and brains but without generating any criticality on a social level. Moreover, dual settings are markedly easier to manage in methodological and technical terms, allowing for the design and implementation of more controlled and standardized studies and therefore the collection of more reliable data. We thus propose the paradigm of C^2^ (C × C): a form of creativity expressed between two individuals that can take full advantage of the opportunities provided by BCI and hyperfeedback (C × C). Indeed, through a dual BCI, it is easy to design creative settings in which two people are engaged in a creative task while each one is wearing a BCI. Each person’s EEG signal is then sent to a computer, and a dedicated software takes care of synchronizing and processing the data so that synchronization, or resonance processes, are detected. The data are then translated into feedback, such as sounds or images, which allows for the recognition of the state in which the dyad is. The system does not necessarily push toward a collaborative phase but rather pushes the dyad to work in a way that also takes advantage of dissonance, which may not be evident on an explicit level (behavior) but only on an implicit level (neuro-cognitive processes). Thus, if one person produces an idea and the other agrees but the feedback signals a divergence, this can be a stimulus for a reformulation of the idea to generate, little by little, a pathway towards real rather than apparent convergence. In this way, C^2^ allows for the full exploitation of the cognitive sharing and potential of a BCI. In particular, when coupled with artificial intelligence engines capable of learning the characteristics of the dyad engaged in the task, it can lead to feedback that is increasingly consistent with the neuro-cognitive configuration of the actual pair. Moving from little-c paradigms to the study of C^2^ allows us to extend the study of creativity to a further level. Indeed, putting two brains to work does not produce a sum of creativities but leads to a non-linear combination that is potentially able to open new opportunities for both research and application. Thus, exploring this frontier of creativity through a BCI will not only further the understanding of the neuro-cognitive processes underlying creative collaboration but will also foster the emergence of plural forms of creativity in real-world contexts, from schools to the enterprise. Furthermore, it will promote hybrid artistic forms that combine the creativity of the dyad, distinct from the creativity of the individual, in collaboration with generative networks and therefore artificial intelligence.

## Figures and Tables

**Figure 1 brainsci-13-00665-f001:**
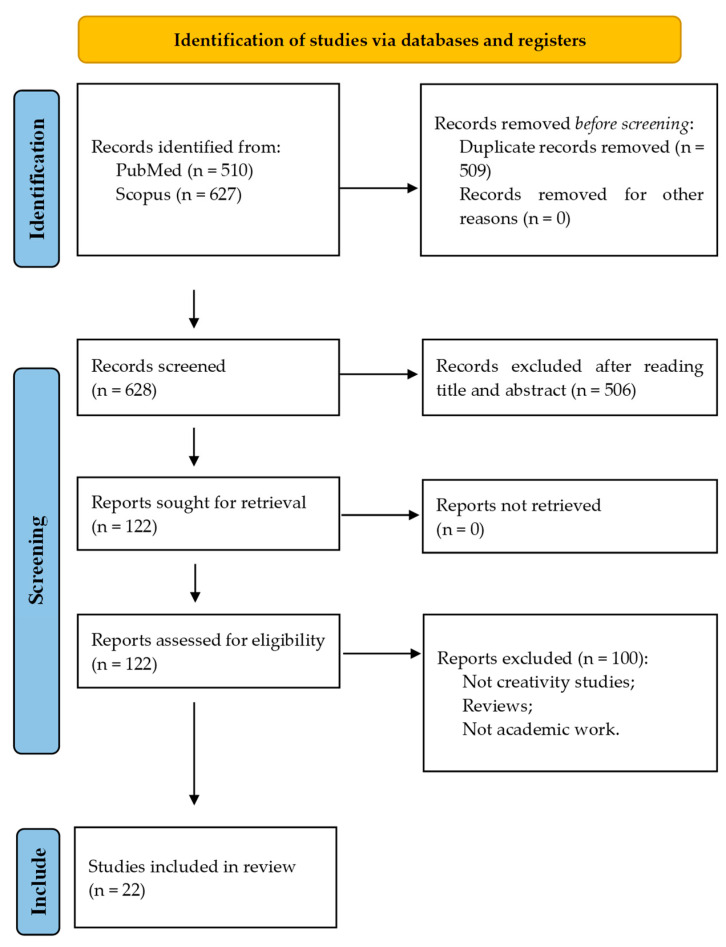
PRISMA flow diagram charting study selection for the scoping review.

**Figure 2 brainsci-13-00665-f002:**
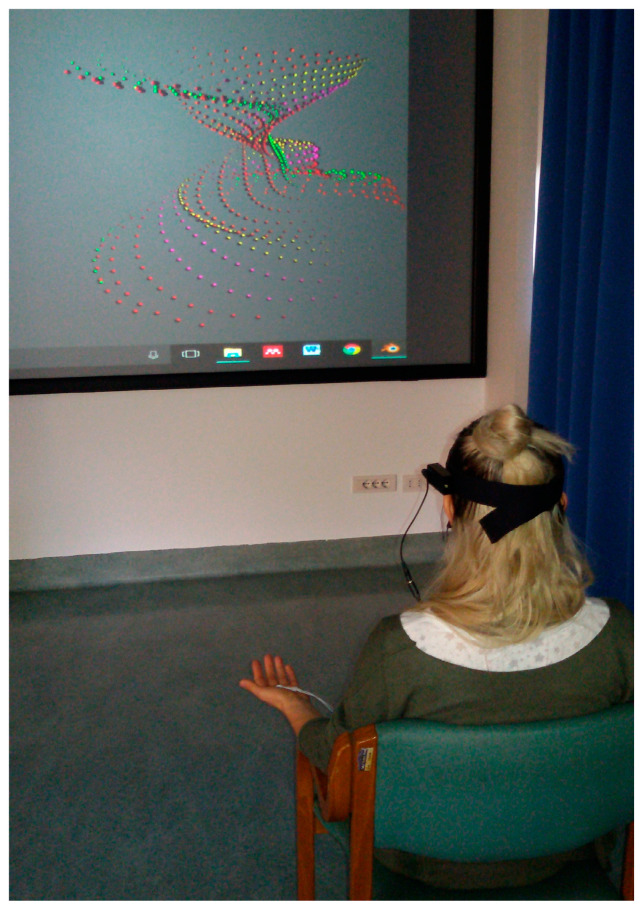
A BCI system controls real-time visualization of DRACLE’s spiral of spheres [32]. Image reproduced with permission of the authors.

**Figure 3 brainsci-13-00665-f003:**
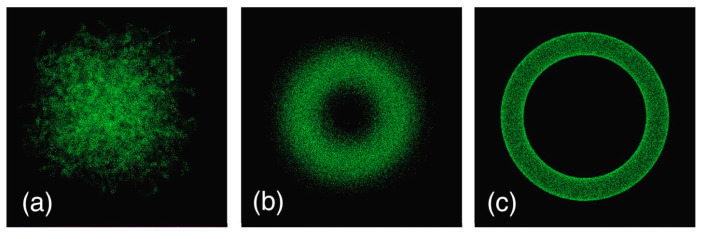
Visualization of the interactive Mandala for (**a**) low (**b**) intermediate, and (**c**) high levels of the meditation (M) parameter (reproduced from [34], page 2—Figure 2, with permission of the authors).

**Table 1 brainsci-13-00665-t001:** Coordinates of the selected papers.

Authors	Year	Country	Sample	Creativity	Assessment	BCI	Main Results
**Clinical applications**							
**Levican et al.** [19]	2018	Chile	N = 1	Music	Music composition	Enobio-8 EEG	-
**Münßinger et al.** [20]	2010	Germany	N = 3 ALS patients N = 10 healthy participants	Painting	Brain Painting	16-channel USBamp	Of patients, 2/3 reached above 89% accuracy.
**Miranda** [21]	2006	UK	-	Music	Music composition	EEG (geodesic net, 19 ch)	-
**Art experience in real-world settings**							
**Pedersen et al.** [22]	2015	Canada	-	Painting	Selection of Klee’s art	Muse	-
**Herrera-Arcos et al.** [23]	2017	Mexico and USA	N = 25	Painting appreciation	Favorite piece	Muse	Suppression of beta while viewing favorite piece over frontal sites.
**BCI for creative content creation**							
*Proto-creativity*							
**Todd et al.** [24]	2012	UK	N = 8	Proto-painting	Painting (free and copy)	EEG with BCI2000 software suite	Task 3 (free drawing) was perceived as the one which allowed for a greater sense of control and was the most enjoyable. Task 2 (copy) was the preferred task.
**Tang et al.** [25]	2022	China	N = 20	Proto-painting	Painting (free and copy)	EEG BioSemi (12 ch)	Hybrid stimulus interface (P300 + SSVEP) was more accurate than P300 alone (88.92%).
**Hamadicharef et al.** [26]	2010	Singapore	-	Proto-music	Music composition	EEG (15 ch)	-
**Pinegger et al.** [27]	2015	Austria	N = 5	Proto-music	Music composition	Mobita EEG system (8 ch)	Three participants reached accuracies above 77% and could produce a given melody.
**Vamvakousis and Ramirez** [28]	2014	Spain	N = 4	Proto-music	Arpeggio shift	Emotiv Epoc (14 ch)	Selection accuracy from 83 to 100%.
*Creative outcomes*							
**Riccio et al.** [29]	2022	Spain, Norway, and Italy	-	Painting	Emotion categorization	EEG database	Happiness, fear, and sadness had the highest sensitivity (>58%) while anger just reached 22% (with a higher sensitivity).
**Kim H.-J. and Kim S.-Y.** [30]	2015	Korea	-	Painting	Self portrait	Neurosky Mindwave	-
**Folgieri and Zichella** [31]	2012	Italy	Task 1: N = 7 Task 2: N = 4	Music	Music composition	Neurosky Mindwave	After a few minutes of training, participants were able to reproduce the notes by thinking of them, with 40–50% immediate success.
**Folgieri et al.** [32]	2017	Italy	-	Sound and visual display	DRACLE	Neurosky Mindwave	-
**Cádiz and de la Cuadra** [33]	2014	Chile	-	Sound and visual display	Multisensorial performance	KARA1: Neurosky Mindwave KARA2: Emotiv Epoc	-
**Tokunaga and Lyons** [34]	2020	Japan	-	Sound and visual display	Mandala	Neurosky Mindwave	-
**Participants’ engagement**							
**Yan et al.** [35]	2016	China	N = 48	Sound and visual display	Adaptive theatre performance	Emotiv Epoc (14 ch)	It is possible to detect significant decreasing thresholds during adaptive theatre performance. There was a better recall of the performance content when using performing cues. The audience was more attracted by multiple performing cues than single performing cues during opera.
**Ramchurn et al.** [36]	2018	UK	-	Movie composition	Brain-controlled movie	Neurosky Mindwave	-
**Ramchurn et al.** [37]	2018	UK	N = 33 questionnaires	Music composition	Musical Soundtracks for BCI Systems	Neurosky Mindwave	The users understood the presence of a relation between the visual elements of the film and the soundtrack.
**Marchesi et al.** [38]	2011	Italy	-	Movie composition	Brain-interactive movie	Neurosky Mindwave	-
**Marchesi** [39]	2012	Italy	-	Correlates of mood during cinema	Video editing	Neurosky Mindwave	-
**Zioga et al.** [40]	2018	UK	N = 7	Live performance	Live performance and video projection	MyndPlay Brain-BandXL	Correlation between the participants’ answers, special elements of the performance, and the audience’s attention, and emotional engagement. The performer’s results were consistent with the recall of representations and the increase in cognitive load.

## Data Availability

No new data were created or analyzed in this study. Data sharing is not applicable to this article.

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
