# Peer review of "BCI Applications to Creativity: Review and Future Directions, from little-c to C^2^"

_brainsci, 2023, doi:10.3390/brainsci13040665_

Round 1
Reviewer 1 Report
The paper provides a systematic overview of the state of the art. The conclusions and outlook on future directions are interesting and sound. However, the novelty of this work is limited - not surpring for a meta-study.
Creativity is an essential concept also for economic innovation and growth. I would have liked to see more in this direction. The study seems to focus mainly on artistic creativity. I was wondering why you didn't add keywords like invention or innovation to your search terms.
I found the paper well structured, well written and easy to understand. However, some sentences are very lengthy and the readability would improve if you used plain English. You might want to engage a native speaker to proof-read.
Author Response
Following the reviewers’ suggestions, we have revised the language in order to increase readability of the text.
Reviewer 2 Report
The authors presented results of a scoping review aimed to describe and systematize the various work that have been conducted on the application potential of the BCI to the field of creativity. The authors rightfully state that the topic is poorly systematized what makes their review a significant contribution to the cutting-edge research area of BCI.
In the introductory part authors present background information: (1) definitions for the creativity and for its basic components: divergent thinking, convergent thinking, flexible cognition, and enactment; (2) localization of the creativity in the brain; based on the appropriate scientific sources.
The review is guided by the research question: "How BCI can provide insight into the creative process?" and conducted within PubMed and Scopus databases, following well-known and suitable PRISMA Statement guideline.
Selection process is properly described and resulted in 22 papers, that were further divided into 4 categories based on the type of creativity they were exploring: clinical applications; art experience in ecological settings; content creations; and participants’ engagement; and the results are presented according this classification. The authors provide description of the experiment setting, study objectives, results and conclusions for each study.
In the concluding part of the paper the authors conclude that : "BCI makes it possible to design and implement studies with much higher ecological validity than traditional studies."
Their conclusion is supported by comprehensive and detailed discussion presenting several advantages and innovations introduced by BCI in research of the complex concept as creativity, emphasizing its dynamic socio-cognitive nature.
Future perspective of the research in the field is identified in multi-brain BCI setting.
Valuable manuscript facilitating future research in this field and providing researchers with comprehensive state of the art.
Excellent and useful paper addressing specific and multidisciplinary topic.
Author Response

(The authors gave the same response as above.)
